# Four New Benzoylamide Derivatives Isolated from the Seeds of *Lepidium apetalum* Willd. and Ameliorated LPS-Induced NRK52e Cells via Nrf2/Keap1 Pathway

**DOI:** 10.3390/molecules27030722

**Published:** 2022-01-22

**Authors:** Meng Li, Beibei Zhang, Mengnan Zeng, Jingke Zhang, Zhiguang Zhang, Weisheng Feng, Xiaoke Zheng

**Affiliations:** 1College of Pharmacy, Henan University of Chinese Medicine, Zhengzhou 450046, China; limeng31716@163.com (M.L.); zhangs9426@163.com (B.Z.); 17320138484@163.com (M.Z.); 18137802812@163.com (J.Z.); 15515881023@163.com (Z.Z.); 2The Engineering and Technology Center for Chinese Medicine Development of Henan Province, Zhengzhou 450046, China

**Keywords:** *Lepidium apetalum* Willd. seeds, benzoylamide derivatives, NRK-52e cell, LPS, Nrf2/Keap1 pathway

## Abstract

Four new benzoylamide derivatives, lepidiumamide B–E (**1**–**4**), were isolated from the seeds of *Lepidium apetalum* Willd. The structures were determined by a combination of MS and NMR analyses. All compounds were evaluated for their protective effects against NRK-52e cell injury induced by lipopolysaccharide (LPS) in vitro. These compounds showed significantly protective activity and ameliorated LPS-induced NRK52e cells via the Nrf2/Keap1 pathway. The discovery of these active compounds is important for the prevention and treatment of renalinjury.

## 1. Introduction

The seeds of *Lepidium apetalum* Willd. have been used as traditional Chinese herbal medicine for the treatment of lung disorders, such as dyspnea, cough, abundant expectoration, and symptoms in which the lung is blocked due to phlegm retention [1]. It was first recorded in Chinese medical classics ‘Shennong’s Herba’ and characterized by its acrid and bitter taste and cold in nature [2]. Furthermore, the dry mature seeds of *L. apetalum* and *Descurainia Sophia* (L.) WebbexPrantl., belonging to Cruciferae (Brassiaceae) family, were commonly called “Tinglizi”, and the former was called “Bei Tinglizi”, and the latter was named “Nan Tinglizi” in the Chinese Pharmacopoeia 2015 [3]. However, they have both been used as “Tinglizi”; there were distinctions between their plant origin, appearance, and pharmacological activity [4]. Compared with the seeds of *L. apetalum*, the previous investigations of *Sophia* D. semen were more comprehensive [5,6,7,8]. Modern pharmacology has demonstrated that the seeds of *L. apetalum* have a cardiotonic [9] effect and the chemical constituents were mainly flavonoids [10,11,12,13]. In our previous report, 11 uridine derivatives were isolated from the seeds of *L. apetalum* [14,15]. As an ongoing study, four new benzoylamide derivatives, lepidiumamide B–E (**1**–**4**) were isolated from the seeds of *L. apetalum* (Figure 1). Herein, the isolation and structural elucidation of the new compounds, as well as their protective effects against NRK-52e cell injury induced by LPS, are described.

## 2. Results and Discussion

### 2.1. Identification of Compounds

Compound **1**, was obtained as pale yellow crystalline powder, had a molecular formula of C_19_H_20_N_2_O_5_ on the basis of [M + Na]^+^ ion at *m/z* 379.1485 (Calcd. 379.1270) in HR-ESI-MS (Appendix A). It showed IR absorptions for hydroxyl (3291 cm^−1^), methyl (2942 cm^−1^), double bond (1639 cm^−1^), and etherlinkage (1230 and 1021cm^−1^) (Appendix A). The ^1^H-NMR spectra (Table 1) of **1** showed the presence of a monosubstituted benzene moiety [*δ*_H_ 7.85 (2H, d, *J* = 7.5 Hz, H-2,6), 7.45 (2H, t, *J* = 7.5 Hz, H-3,5), and 7.52 (1H, m, H-4)] and a *p*-substituted benzene moiety [*δ*_H_ 7.07 (2H, d, *J* = 8.5 Hz, H-2′,6′) and 6.67 (2H, d, *J* = 8.5 Hz, H-3′,5′)] [16]. Furthermore, two methylene groups [*δ*_H_ 2.40 (2H, m, H-10), 2.31 (1H, m, H-11a), and 2.14 (1H, m, H-11b)] and a methine [*δ*_H_ 4.59 (1H, m, H-9)] revealed the presence of the moiety of -NH-CH-CH_2_-CH_2_-. The ^13^C NMR spectrum (Table 1) indicated 15 carbon resonances, including three carbonyls [*δ*_C_ 175.0 (C-15), 174.7 (C-12), and 170.3 (C-7)], a monosubstituted benzene moiety carbonyls [*δ*_C_ 130.5 (C-1′), 130.0 (C-2′,6′), 116.2 (C-3′,5′), and 157.7 (C-4′)], a *p*-substituted benzene moiety carbonyls [*δ*_C_ 135.1 (C-1), 128.5 (C-2,6), 129.5 (C-3,5), and 130.9 (C-4)], 3 methylenes [*δ*_C_ 43.8 (C-14), 33.5 (C-11), and 28.2 (C-10)], and a methine [*δ*_C_ 54.0 (C-9)]. These spectral data suggested the presence of a benzoylamide derivative [17,18]. Our assignments were supported by HMBC data (Appendix A), which showed correlations from *δ*_H_ 4.22 (H-14) to 130.5(C-1′) and 174.7 (C-12), from 2.31 (H-11a) and 2.14 (H-11b) to 33.5 (C-11), 54.0 (C-9), and 174.7 (C-12), from 4.59 (H-9) to 170.3 (C-7) and 175.0 (C-15), and from 7.85 (H-2,6) to 170.3 (C-7) (Figure 2). All the protons and carbons were unambiguously assigned (Table 1) by ^1^H-^1^H COSY (Appendix A), HSQC (Appendix A), and HMBC (Appendix A) experiments. Thus, the structure of Compound **1** was elucidated as *N*^2^-benzoyl-*N*^5^-(4-hydroxybenzyl) glutamine and named lepidiumamide B (Figure 1).

Compound **2** was isolated as a pale-yellow crystalline powder. Its molecular formula was determined to be C_20_H_22_N_2_O_5_ on the basis of the HRESIMS (*m/z* 393.1412 [M + Na]+; calcd for C_20_H_22_N_2_O_5_Na 393.1426) (Appendix A) and the ^13^C NMR data (Appendix A). The ^1^H NMR data (Table 1) displayed a monosubstituted benzene moiety [*δ*_H_ 7.85 (2H, d, *J* = 8.5 Hz, H-2,6), 7.45 (2H, t, *J* = 8.5 Hz, H-3,5), 7.54 (1H, m, H-4)] and a *p*-substituted benzene moiety [*δ*_H_ 7.07 (2H, d, *J* = 8.5 Hz, H-2,6) and 6.69 (2H, d, *J* = 8.5 Hz, H-2,6)], a methine [*δ*_H_ 4.59 (1H, m, H-9)], a methoxy group 3.73 (3H, s, -OCH_3_), and two methylene groups [*δ*_H_ 2.40 (2H, m, H-10), 2.31 (1H, m, H-11a) and 2.14 (1H, m, H-11b)]. The ^13^C-NMR (Table 2) showed 16 carbon signals, which was confirmed by the DEPT spectra (Appendix A) and HSQC (Appendix A) experiments to be a·methoxy group, 3 methylenes, 6 methines, and 6 quaternary carbons. The structure of compound **2** was very similar to compound **1**, except one proton attached with C-15 was replaced by one methoxy group [*δ*_C_ 52.8 (-OCH_3_)] [19]. This finding was supported by the HMBC correlation from *δ*_H_ 3.73 to *δ*_C_ 173.8 (C-15) (Figure 2). On the basis of the above analysis, the structure of **2** was determined as methyl *N*^2^-benzoyl-*N*^5^-(4-hydroxybenzyl) glutamine and was named lepidiumamide C (Figure 1).

Compound **3** was isolated as a pale-yellow crystalline powder. The HRESIMS ion at 377.1704 [M + Na]^+^ (Appendix A) and ^13^C NMR data (Appendix A) of **3** established its molecular formula as C_20_H_22_N_2_O_4_. The NMR spectra exhibited the signals of two monosubstituted benzene moieties [*δ*_H_ 7.88 (2H, d, *J* = 8.0 Hz, H-2,6)/*δ*_C_ 127.4 (C-2,6), *δ*_H_ 7.47 (2H, m, H-3,5)/*δ*_C_ 128.2 (C-3,5), *δ*_H_ 7.54 (1H, m, H-4)/*δ*_C_ 131.5 (C-4); *δ*_H_ 7.28 (2H, m, H-3′,5′)/*δ*_C_ 128.2 (C-3′,5′), *δ*_H_ 7.21 (3H, m, H-2′,4′,6′)/*δ*_C_ 127.1 (C-2′,6′), 126.7 (C-4′)]. The NMR data of **3** (Table 1) closely resembled those of **2**. The only remarkable difference was the substitution type of the benzene moiety [*δ*_H_ 7.28 (2H, m, H-3′,5′), 7.21 (3H, m, H-2′,4′,6′); *δ*_C_ 128.2 (C-3′,5′), 127.1 (C-2′,6′), 126.7 (C-4′)], which was a monosubstituted benzene moiety in **3** instead of the *p*-substituted benzene moiety in **2** at C-14 [20]. Accordingly, compound **3** was determined as methyl *N*^2^-benzoyl-*N*^5^-benzylglutaminate and named lepidiumamide D (Figure 1).

Compound **4** was isolated as a light yellow, amorphous powder. The molecular formula C_18_H_20_N_2_O_4_ was established by HRESIMS at *m/z* 351.1413 [M + Na]^+^ (Appendix A) and ^13^C NMR data (Appendix A). The ^1^H NMR data of **4** (Table 2) indicated the presence of an ABX pattern [*δ*_H_ 7.43 (1H, d, *J* = 2.0 Hz, H-2), 6.83 (1H, d, *J* = 8.0 Hz, H-5), and 7.41 (1H, dd, *J* = 2.0, 8.0 Hz, H-6)] [21], a monosubstituted benzene moiety [*δ*_H_ 7.39 (2H, m, H-3′,5′) and 7.32 (3H, m, H-2′,4′,6′)], three methylene moieties [*δ*_H_ 4.52 (2H, s, H-13), 3.92 (2H, t, *J* = 8.0 Hz, H-9), and 3.55 (2H, t, *J* = 8.0 Hz, H-10)], and a methoxy group [*δ*_H_ 3.79 (3H, s, -OCH_3_)]. The ^13^C NMR spectroscopic data (Table 2) of **4** displayed resonance for 16 carbons, which was confirmed by the DEPT (Appendix A) and HSQC (Appendix A) experiments to be a·methoxy group [*δ*_C_ 55.5 (-OCH_3_)], 3 methylenes [*δ*_C_ 49.8 (C-9), 48.2 (C-13), and 31.1 (C-10)], 6 methines, and 6 quaternary carbons. The two quaternary carbon signals at *δ*_C_ 167.2 (C-11) and 163.9 (C-7) indicated the presence of two carbonyl groups (amide groups). Six downfield carbon signals at *δ*_C_ 121.6 (C-1), 112.7 (C-2), 151.1 (C-3), 147.2 (C-4), 115.0 (C-5), and 123.4 (C-6) were assignable to olefinic carbon atoms as members of the ABX system, and four downfield carbon signals at *δ*_C_ 135.8 (C-1′), 127.7 (C-2′,6′), 128.7 (C-3′,5′), and 128.0 (C-4′) were the monosubstituted benzene moiety. In the ^13^C NMR spectrum (Appendix A), the signals of *δ*_C_ 163.9 (C-7), 49.8 (C-9), and 48.2 (C-13) observedwere particularly low, suggesting that these three carbon atoms may be connected with the N atom. These spectral data suggested the presence of a benzoylamide derivative. The HMBC spectrum (Appendix A) indicated correlations of the H-2/6 and C-7, H-9 and C-7/10, and H-13 and C-11/1′ (Figure 2). Based on the above spectroscopic data, compound **4** was characterized as *N*-[3-(benzylamino)-3-oxopropyl]-4-hydroxy-3-methoxy benzamide and named lepidiumamide E (Figure 1).

### 2.2. Effects of Compounds ***1***–***4*** on LPS-Induced NRK-52e Cell Viability

As shown in Figure 3, the NRK-52e cell viability was reduced by LPS, and compounds **1**–**4** could effectively enhance the cell viability of NRK-52e induced by LPS.

### 2.3. Effects of Compounds ***1***–***4*** on the Levels of Keap1 and p-Nrf2 in NRK-52e Cell Induced by LPS

As shown in Figure 4, the level of Keap1 was increased and the level of p-Nrf2 was decreased in LPS-induced NRK-52e cells, which could be reversed by compounds **1**–**4**.

## 3. Materials and Methods

### 3.1. General Experimental Procedures

NMR spectra were recorded at room temperature on a Bruker Avance III 500 MHz spectrometer (Bruker Biospin, Rheinstetten, Germany). Optical rotations were performed on an APIV (Rudolph Research Analytical, Hackettstown, NJ, USA). IR spectroscopy was determined on a Nicolet iS10 Microscope Spectrometer (Thermo Fisher Scientific, Waltham, MA, USA). HRESIMS spectra were obtained on a Bruker maXis HD mass spectrometer. UV spectra were obtained on a Shimadzu UV-2401PC apparatus (Shimadzu, Kyoto, Japan). Preparative HPLC was performed on a Saipuruisi LC-50 instrument with a UV200 detector and a YMC-Pack ODS-A column (250 × 20 mm, 5 μm and 250 × 10 mm, 5 μm, Saipuruisi, Beijing, China). Silica gel (200–300 mesh, Qingdao Marine Chemical Inc., Qingdao, China), LiChroprep RP-C18 gel (40–63 μm, Merck, Germany), Diaion HP-20 (Mitsubishi Chemical Corporation, Shanghai, China),Toyopearl HW-40, MCI gel CHP-20 (TOSOH Corporation, Tokyo, Japan), and Sephadex LH-20 (Cytiva Sweden AB, Uppsala, Sweden) were used for open column chromatography (CC). The organic solvents we used in the plant material extract and column chromatography were all of analytical grade, and the organic solvents used in semi-preparative HPLC were HPLC-grade, which were supplied by Tianjin Fuyu Fine Chemical Co., Ltd. (Tianjin, China).

Dulbecco’s Modified Eagle Medium and phenol-red-free Dulbecco’s Modified Eagle’s Medium (Gibco, Pittsburgh, PA, USA); Heat-Inactivated Fetal Bovine Serum (HyClone, Logan, UT, USA); 17*β*-E2 (positive drugs for cell experimentation, Sigma, Louis, MO, USA); methyl thiazolyl tetrazolium (MTT, Beijing Solarbio Technology Co., Ltd.) and dimethyl sulfoxide (DMSO) (Amresco, Seattle, WA, USA); MPP (Tocris, Bristol, UK); Odyssey Infra-red Imager (LI-COR Biosciences, Lincoln, NE, USA); Microplate reader (Bio-Rad, Hercules, CA, USA); CO_2_ incubator (Thermo Scientific, Waltham, MA, USA).

### 3.2. Plant Materials

The seeds of *L*. Willd. (Brassicaceae) were collected in June 2014 from Taiping town, Xixia county, Nanyang city, Henan province, China (33°37′N, 111°44′E). The plants were identified by Professors Suiqing Chen and Chengming Dong of Henan University of Chinese Medicine. A voucher specimen (No. 20141101A) has been deposited in the Department of Natural Medicinal Chemistry, School of Pharmacy, Henan University of Chinese Medicine, Zhengzhou, China.

### 3.3. Extraction and Isolation

The processed seeds of *L. apetalum* (8.0 kg) were extracted three times with H_2_O (80 L×3, 1.5 h each time) at 100 °C. Evaporation of the solvent under reduced pressure provided aqueous extracts (1.04 kg); it was then precipitated at an ethanol concentration of 80% (4 L×3), and the liquid supernatant was concentrated in a vacuum evaporator to yield a gross extract (628 g), which was suspended in H_2_O (1.5 L). The extract was put through a Diaion HP-20 macroporous resin column and eluted with EtOH-H_2_O (0:100→60:40), successively, to obtain four fractions (A–D), respectively.

Fraction D (67 g) was suspended in water, the dissoluble part was marked as fraction DA, and the insoluble part as fraction DB. Fraction DB was applied to a silica gel column and eluted with CH_2_Cl_2_-MeOH (0:100→1:1) to yield 5 fractions (Fr.1–Fr.5). Fraction 4 was separated using an open ODS column (0%→60%, *v*/*v*) to afford 6 fractions (Fr.4.1–4.6). Fr.4.4 was further purified by semipreparative HPLC (CH_3_CN: 0.03% CF_3_COOH-H_2_O, 27:73, 3 mL/min) to obtain Compound **1** (10.1 mg, t_R_ = 36.1 min). Fr.4.5 was separated by Toyopearl HW-40 CC (MeOH, 100%) to yield Fr.4.5.1–4.5.4. Fr4.5.1 was subjected to semi-preparative HPLC (CH_3_CN: 0.03% CF_3_COOH-H_2_O, 11:89, 3 mL/min) to obtain Compound **4** (1.29 mg, t_R_ 36.1 min). Fr4.5.2 was subjected to semi-preparative HPLC (CH_3_CN: 0.03% CF_3_COOH-H_2_O, 20:80, 3 mL/min) to obtain Compound **2** (2.0 mg, t_R_ 66.7 min). Fr4.5.4 was subjected to semi-preparative HPLC (CH_3_CN: 0.03% CF_3_COOH-H_2_O, 35:65, 3 mL/min) to obtain Compound **3** (3.59 mg, t_R_ 27.6 min).

### 3.4. Compound Characterization

Lepidiumamide B (**1**): Pale yellow crystalline powder; [α${]}_{D}^{20}$ = −2.0 (c 0.10, MeOD); UV (MeOH) λ_max_(log ε) nm: 204 (2.47), 225 (2.14), 275 (0.36); IR *ν*_max_cm^−^^1^(iTR): 3291, 2942, 1639, 1234, 1021, 712 cm^−^^1^; ^1^H NMR (500 MHz, MeOD, see Table 1) spectral data and ^13^C NMR (125 MHz, MeOD, see Table 1) spectral data, HR-ESI-MS *m/z* 379.1249 [M + Na]^+^ (calcd for C_19_H_20_N_2_O_5_Na 379.1270).

Lepidiumamide C (**2**): Pale yellow crystalline powder; [α${]}_{D}^{20}$= −4.8 (c 0.02, MeOD); UV (MeOH) λ_max_(log ε) nm: 204 (2.74), 224 (1.93), 279 (0.66) nm; IR *ν*_max_cm^−^^1^(iTR): 3308, 2930, 2856, 1641, 1613, 1516, 1202, 1027,720 cm^−^^1^; ^1^H NMR (500 MHz, MeOD, see Table 1) spectral data and ^13^C NMR (125 MHz, MeOD, see Table 1) spectral data, HR-ESI-MS *m/z* 393.1412 [M + Na]^+^ (calcd for C_20_H_22_N_2_O_4_Na 393.1426).

Lepidiumamide D (**3**): Pale yellow crystalline powder; [α${]}_{D}^{20}$ = −2.9 (c 0.04, MeOD); UV (MeOH) λ_max_(log ε) nm: 205 (2.42), 227 (1.27); IR *ν*_max_cm^−^^1^(iTR): 3300, 3064, 2929, 1642, 1536, 1437, 1205, 698 cm^−^^1^; ^1^H NMR (500 MHz, DMSO-*d*_6_, see Table 1) spectral data and ^13^C NMR (125 MHz, DMSO-*d*_6_, see Table 1) spectral data, HR-ESI-MS *m/z* 377.1464 [M + Na]^+^ (calcd for C_20_H_22_N_2_O_4_Na 377.1477).

Lepidiumamide E (**4**): Pale yellow crystalline powder; [α${]}_{D}^{20}$ = −4.4 (c 0.01, MeOD); UV (MeOH) λ_max_(log ε) nm: 208 (1.28), 215 (1.25), 260 (0.42), 295 (0.25); IR *ν*_max_cm^−^^1^(iTR): 3160, 2931, 2853, 1675, 1623, 1184, 1133, 721cm^−^^1^; ^1^H NMR (500 MHz, DMSO-*d*_6_, see Table 1) spectral data and ^13^C NMR (125 MHz, DMSO-*d*_6_, see Table 2) spectral data, HR-ESI-MS *m/z* 351.1413 [M + Na]^+^ (calcd for C_18_H_20_N_2_O_4_Na 351.1321).

### 3.5. Cell Culture and MTT Assay

The NRK-52e cell line was purchased from ATCC (MD, USA) and maintained in DMEM supplemented with 10% FBS and antibiotics (100 IU/mL penicillin and 100 mg/mL streptomycin) in a humidified atmosphere of 5% CO_2_ at 37 °C.

The NRK-52e cells were divided into seven groups: control (CON), model (LPS, 1 μg/mL), Vitamin E (VE, 10μM + LPS, 1 μg/mL, positive control), and compounds **1**–**4** (10μM + LPS, 1 μg/mL). Twenty-four hours later, the cell viability was detected by the MTT assay. An independent parallel test was repeated three times [22].

### 3.6. Immunofluorescence

The NRK-52e cells were fixed with 4% paraformaldehyde for 15 min and washed with ice-cold PBS. Then, they were incubated with 0.25% Triton X-100 in PBS for 10 min and rinsed three times with PBS for 5 min. For immunofluorescence staining, the cells were blocked in PBST with 2% bovine serum albumin (BSA, Genview; Beijing, China) for 30 min, followed by incubation with primary antibodies Keap1 (ab139729, Abcam, MA, USA) and p-Nrf2 (ab76026, Abcam) overnight at 4 °C. After washing with PBS, the cells were stained with Alexa Fluor^TM^ 594 donkey anti-rabbit IgG (H + L) (1981155, Thermofisher) for 1 h in the dark and rinsed 3 times with PBS for 5 min. The cells were then incubated with DAPI for 3 min and then rinsed with PBS. Finally, the plate was scanned using Cytation 5 (BioTek, Shoreline, WA, USA) [23].

### 3.7. Statistical Analysis

Data were analyzed using SPSS 26.0 software (IBM; Endicott, NY, USA). Statistical significance was assessed in comparison with the respective control for each experiment using one-way analysis of variance. A *p* value less than 0.05 indicated a statistically significant difference.

## 4. Conclusions

In this study, four new benzoylamide derivatives, lepidiumamide B–E (**1**–**4**) were isolated from the seeds of *L. apetalum*. The structures were determined by a combination of MS and NMR analyses. The isolated compounds were evaluated for their protective effects against NRK-52e cell injury induced by LPS in vitro and showed significantly protective activity, as well asameliorated LPS-induced NRK52e cells via the Nrf2/Keap1 pathway.

## 5. Patents

There wasa patent resulting from the work reported in this manuscript: Tinglixinan D andTinglixinan E and their preparation method and application (ZL 201710796478.3).The patentee was Henan University of Chinese Medicine, and the patent inventorswereFeng, Weisheng; Li, Meng; Zeng, Mengnan; Zheng, Xiaoke; Zhang, Jingke; Zhao, Xuan; Zhang, Zhiguang; and Lv, Jinjin.

## Figures and Tables

**Figure 1 molecules-27-00722-f001:**
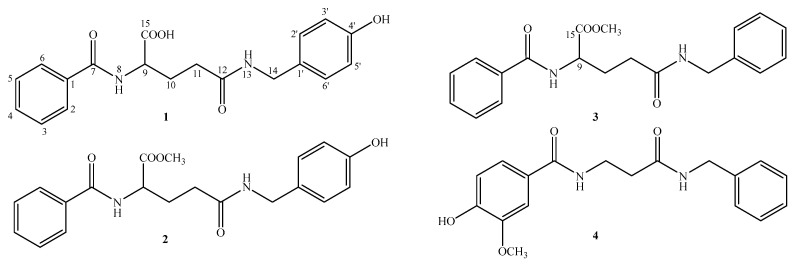
Structures of compounds **1**–**4** from *L. apetalum*.

**Figure 2 molecules-27-00722-f002:**
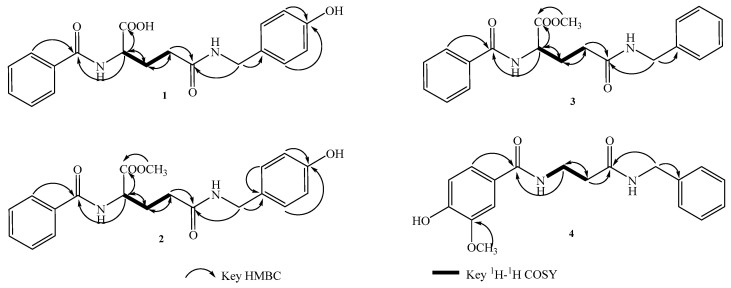
Key HMBC and ^1^H-^1^H COSY correlations of compounds **1**–**4**.

**Figure 3 molecules-27-00722-f003:**
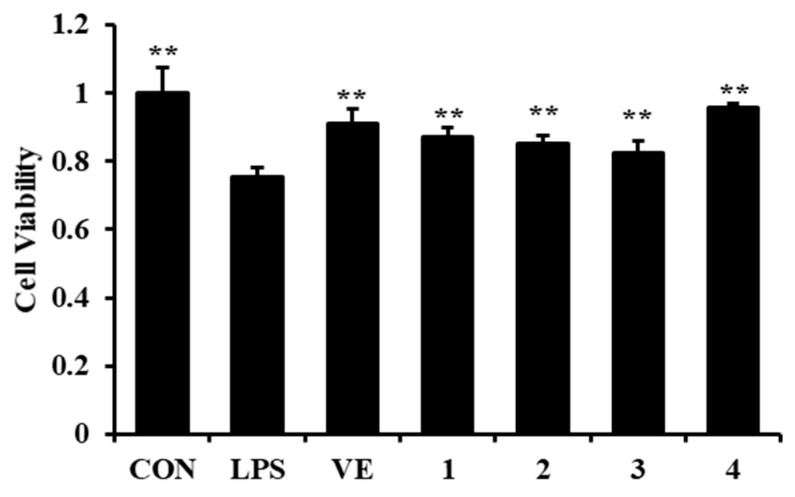
The effects of compounds **1**–**4** on LPS-induced NRK-52e cell viability. Data are expressed as *x ± sd*, *n* = 3, * *p* < 0.05, ** *p* < 0.01 compared with the LPS group.

**Figure 4 molecules-27-00722-f004:**
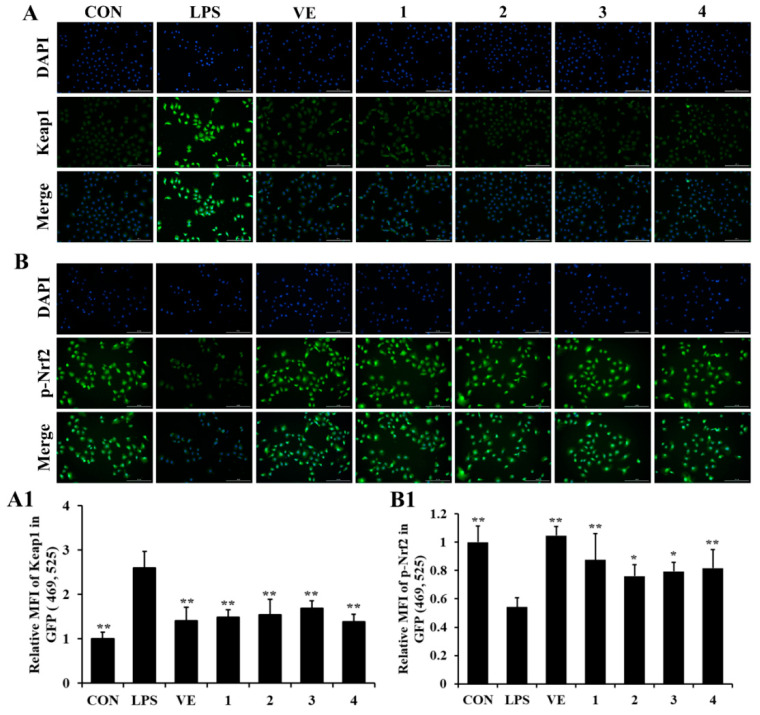
The effects of compounds **1**–**4** on the levels of Keap1 and p-Nrf2 in NRK-52e cell induced by LPS. Representative plots showing the effects of compounds **1**–**4** on Keap1 (**A**) and p-Nrf2 (**B**) detected by immunofluorescence. Quantitative results of compounds **1**–**4** on Keap1 (**A1**) and p-Nrf2 (**B1**) detected by immunofluorescence. Data are expressed as *x ± sd*, *n* = 3, * *p* < 0.05, ** *p* < 0.01 compared with the LPS group.

**Table 1 molecules-27-00722-t001:** ^1^H-(500 MHz) and ^13^C-NMR (125 MHz) data for compounds **1**–**3**.

No.	1 ^a^		2 ^a^		3 ^b^	
	*δ* _H_	*δ* _C_	*δ* _H_	*δ* _C_	*δ* _H_	*δ* _C_
1		135.1		135.0		133.7
2,6	7.85 (2H, d, 7.5)	128.5	7.85 (2H, d, 7.5)	128.5	7.88 (2H, d, 8.0)	127.4
3,5	7.45 (2H, t, 7.5)	129.5	7.45 (2H, t, 7.5)	129.6	7.47 (2H, t, 7.5)	128.2
4	7.52 (1H, m)	132.9	7.54 (1H, m)	133.0	7.54 (1H, m)	131.5
7		170.3		170.3		166.5
8					8.80 (1H, d, 7.0)	
9	4.59 (1H, m)	54.0	4.59 (1H, m)	54.2	4.59 (1H, m)	52.4
10	2.31 (1H, m) 2.14 (1H, m)	28.2	2.30 (1H, m) 2.14 (1H, m)	28.0	2.40 (2H, m)	26.3
11	2.40 (2H, m)	33.5	2.40 (2H, m)	33.3	2.31 (1H, m) 2.14 (1H, m)	31.6
12		174.7		174.5		171.3
13					8.36 (1H, t, 6.0)	
14	4.22 (2H, d, 4.0)	43.8	4.22 (2H, d, 4.0)	43.8	4.22 (2H, d, 4.0)	42.0
15		175.0		173.8		172.5
1′		130.5		130.5		139.4
2′,6′	7.07 (2H, d, 8.5)	130.0	7.07 (2H, d, 8.5)	130.0	7.21 (2H, d, 8.5)	127.1
3′,5′	6.67 (2H, d, 8.5)	116.2	6.69 (2H, d, 8.5)	116.2	7.28 (2H, d, 8.5)	128.2
4′		157.7		157.7	7.21 (2H, d, 8.5)	126.7
OCH_3_			3.73 (3H, s)	52.8	3.64 (3H, s)	51.8

^a^ is recorded in CD_3_OD; ^b^ is recorded in DMSO-*d*_6_

**Table 2 molecules-27-00722-t002:** ^1^H-(500 MHz) and ^13^C-NMR (125 MHz) data for compound **4** (in DMSO-*d*_6_).

No.	*δ* _H_	*δ* _C_		*δ* _H_	*δ* _C_
1		121.6	10	3.55 (2H, t, 8.0)	31.1
2	7.43 (1H, d, 2.0)	112.7	11		167.2
3		151.1	13	4.52 (2H, s)	48.2
4		147.2	1′		135.8
5	6.83 (1H, d, 8.5)	115.0	2′,6′	7.32 (2H, m)	127.7
6	7.41 (1H, dd, 2.0, 8.5)	123.3	3′,5′	7.39 (2H, m)	128.7
7		163.9	4′	7.32 (1H, m)	128.0
9	3.92 (2H, t, 8.0)	49.8	OCH_3_	3.79 (3H, s)	55.5

## Data Availability

Data are contained within the manuscript.

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
