# Peer review of "Four New Benzoylamide Derivatives Isolated from the Seeds of Lepidium apetalum Willd. and Ameliorated LPS-Induced NRK52e Cells via Nrf2/Keap1 Pathway"

_molecules, 2022, doi:10.3390/molecules27030722_

Round 1

Reviewer 1 Report

The main importance of the manuscript is the procedure of isolation of several new compounds from the seeds of Lepidium apetalum and their detailed structure elucidation. Additionally, some biological activity has been determined that could be important for the prevention and treatment of renal injury. The work is well presented and needs only minor revision before publication as described below:

  • Some abbreviations should be presented when first mentioned, e, g. LBS (p.1 r. 42);
  • p. 2 r.61 – "methine" instead of "methane";
  • In Tables 1 and 2 the chemical shifts of the detected NH protons should be listed;
  • p. 4, r. 121 – The sentence “were particularly low, suggesting that these three carbons may be connected with N atoms” is not correct, it should be revised;
  • p. 6, r. 148 - "IR spectroscopy" would be better than "IR spectrometry";
  • p. 7, rows 201, 206, 211, 216 the table indications should be corrected;
  • p. 9, r.307 In reference 21 please, omit ().

Author Response

Dear Editor:

Thank you very much for your letter and the comments about our paper submitted to journal of Molecules (molecules-1570830). The manuscript entitled “Four new benzoylamide derivatives isolated from the seeds of Lepidium apetalum Willd. and ameliorated LPS-induced NRK52e cells via Nrf2/Keap1 pathway” by Meng Li, Beibei Zhang, Mengnan Zeng, Jingke Zhang, Zhiguang Zhang, Wei-sheng Feng, Xiao-ke Zheng have been revised according to the reviewers’ comments, and we wish it to be reconsidered for publication in Journal of Molecules.

A list of changes and responses to reviewers are as follows.

  1. Some abbreviations should be presented when first mentioned, e, g. LBS (p.1 r. 42);

Answer: LPS in the part of Abstract had been revised as "lipopolysaccharide (LPS)".

  1. p. 2 r.61 – "methine" instead of "methane";

Answer: "methine" had been revised as "methane".

  1. In Tables 1 and 2 the chemical shifts of the detected NH protons should be listed;

Answer: In Table 1, the 1H NMR spectrum of compounds 1 and 2 in CD3OD can't be detected the NH protons, but the 1H NMR spectrum of compound 3 in DMSO-d6 showed the presence of NH protons at d H 8.80 (1H, d, J = 7.0 Hz, NH-7) and 8.36 (1H, t, J = 6.0 Hz, NH-13), which were listed in Table 1. The 1H NMR spectrum of compound 4 was in DMSO-d6 also, but the signals of NH protons can't be observed.

  1. p. 4, r. 121 – The sentence “were particularly low, suggesting that these three carbons may be connected with N atoms” is not correct, it should be revised;

Answer: The sentence “were particularly low, suggesting that these three carbons may be connected with N atoms” had been revised as “the signals of d C 163.9 (C-7), 49.8 (C-9) and 48.2 (C-13) observed were particularly low, suggesting that these three carbon atoms may be connected with N atom. ”

  1. p. 6, r. 148 - "IR spectroscopy" would be better than "IR spectrometry";

Answer: "IR spectrometry" had been revised as "IR spectroscopy"

  1. p. 7, rows 201, 206, 211, 216 the table indications should be corrected;

Answer: The table indications had been revised.

  1. p. 9, r.307 In reference 21 please, omit ().

Answer: "()" in reference 21 had been deleted.

All in all, thank you very much for your reconsidering our revised manuscript for potential publication in Phytochemistry. I'm looking forward to hearing from you soon. Correspondence should be addressed to Weisheng Feng at the following address, phone and fax number, and email address.

Best Regards.

Yours Sincerely,

Meng Li

Tel./Fax.: + 8637160190296

E-mail: fwsh@hactcm.edu.cn

Jinshui east Road,

Henan University of Chinese Medicine

Zhengzhou 450046,

People's Republic of China.

Reviewer 2 Report

Dear Authors,

This manuscript described the discovery of four new metabolites from the seeds of Lepidium apetalum Willd. collected at China. The isolated compounds were tested against LPS-induced NRK52e cells.

1) Figure 4, the caption for A1 and B1 are missing.

2) Configuration at C-9 in 1-3 is not determined. Because it might be present in a racemic mixture, enantiomer, and caused the optical rotation to be near to zero. 

3) Was lepidiumamide A using CD3OD for NMR measurement? Because NH signal is not able to observed in this solvent. Therefore, position 12 can be exchanged with 13, while it didn't compromise the HMBC data. Of course, looking at chemical shifts of C-14, we can deduced the structure is most likely as Authors have proposed.  

4) Figure 2, COSY correlation should be depicted as bold line.

5) Is there a possibility that the O-methylation on COOH in 2 an artifact?

6) All the important spectra such as NMR and MS in supplementary materials are not in a good quality, it is very blurred.

Author Response

Dear Editor:

Thank you very much for your letter and the comments about our paper submitted to journal of Molecules (molecules-1570830). The manuscript entitled “Four new benzoylamide derivatives isolated from the seeds of Lepidium apetalum Willd. and ameliorated LPS-induced NRK52e cells via Nrf2/Keap1 pathway” by Meng Li, Beibei Zhang, Mengnan Zeng, Jingke Zhang, Zhiguang Zhang, Wei-sheng Feng, Xiao-ke Zheng have been revised according to the reviewers’ comments, and we wish it to be reconsidered for publication in Journal of Molecules.

A list of changes and responses to reviewers are as follows.

1) Figure 4, the caption for A1 and B1 are missing.

Answer: The caption for A1 and B1 of Figure 4 had been supplemented.

2) Configuration at C-9 in 1-3 is not determined. Because it might be present in a racemic mixture, enantiomer, and caused the optical rotation to be near to zero.

Answer: The solution to this question is to do chiral separation for compounds. Unfortunately, the amount of the compounds is too small to complete the chiral separation. It's also the deficiency of our experiment.

3) Was lepidiumamide A using CD3OD for NMR measurement? Because NH signal is not able to observed in this solvent. Therefore, position 12 can be exchanged with 13, while it didn't compromise the HMBC data. Of course, looking at chemical shifts of C-14, we can deduced the structure is most likely as Authors have proposed.

Answer: Solubility and sample recovery were considered for the selection of solvents, therefore Lepidiumamide A and B were using CD3OD for NMR measurement. Although NH signal is not able to observed in this solvent, the structures of the compounds were also determined.

4) Figure 2, COSY correlation should be depicted as bold line.

Answer: COSY correlation in Figure 2 had been depicted as bold line.

5) Is there a possibility that the O-methylation on COOH in 2 an artifact?

Answer: The question is uncertain. Many compounds contain methoxyl groups in natural products. In addition, the low content of this compound makes it difficult to determine whether it is native to L. apetalum by HPLC or UPLC-MS.

6) All the important spectra such as NMR and MS in supplementary materials are not in a good quality, it is very blurred.

Answer: All the important spectra such as NMR and MS in supplementary materials are not in a good quality, but it does not affect the analysis of these spectra.

All in all, thank you very much for your reconsidering our revised manuscript for potential publication in Molecules. I'm looking forward to hearing from you soon. Correspondence should be addressed to Weisheng Feng at the following address, phone and fax number, and email address.

Best Regards.

Yours Sincerely,

Meng Li

Tel./Fax.: + 8637160190296

E-mail: fwsh@hactcm.edu.cn

Jinshui east Road,

Henan University of Chinese Medicine

Zhengzhou 450046,

People's Republic of China.

Round 2

Reviewer 2 Report

The major concerns have been addressed positively. I have no further comments.